# Gut Microbiome Composition and Its Metabolites Are a Key Regulating Factor for Malignant Transformation, Metastasis and Antitumor Immunity

**DOI:** 10.3390/ijms24065978

**Published:** 2023-03-22

**Authors:** Stefan Lozenov, Boris Krastev, Georgi Nikolaev, Monika Peshevska-Sekulovska, Milena Peruhova, Tsvetelina Velikova

**Affiliations:** 1Laboratory for Control and Monitoring of the Antibiotic Resistance, National Centre for Infectious and Parasitic Diseases, 26 Yanko Sakazov Blvd, 1504 Sofia, Bulgaria; dr_lozenov@abv.bg; 2Nadezhda Paradise Medical Center, 1330 Sofia, Bulgaria; bmk83@abv.bg; 3Department of Cell and Developmental Biology, Faculty of Biology, Sofia University “St. Kliment Ohridski”, 1504 Sofia, Bulgaria; 4Department of Gastroenterology, University Hospital Lozenetz, Sofia, Medical Faculty, Sofia University “St. Kliment Ohridski”, 1407 Sofia, Bulgaria; 5Department of Gastroenterology, University Hospital Heart and Brain, 5804 Pleven, Bulgaria; 6Medical Faculty, Sofia University St. Kliment Ohridski, Kozyak 1 str., 1407 Sofia, Bulgaria

**Keywords:** microbiome, gut microbiota, oncogenesis, carcinogenesis, colorectal cancer, interleukin pathways, NF-kB, Th17 cells, tumor suppression, HDACs

## Abstract

The genetic and metabolomic abundance of the microbiome exemplifies that the microbiome comprises a more extensive set of genes than the entire human genome, which justifies the numerous metabolic and immunological interactions between the gut microbiota, macroorganisms and immune processes. These interactions have local and systemic impacts that can influence the pathological process of carcinogenesis. The latter can be promoted, enhanced or inhibited by the interactions between the microbiota and the host. This review aimed to present evidence that interactions between the host and the gut microbiota might be a significant exogenic factor for cancer predisposition. It is beyond doubt that the cross-talk between microbiota and the host cells in terms of epigenetic modifications can regulate gene expression patterns and influence cell fate in both beneficial and adverse directions for the host’s health. Furthermore, bacterial metabolites could shift pro- and anti-tumor processes in one direction or another. However, the exact mechanisms behind these interactions are elusive and require large-scale omics studies to better understand and possibly discover new therapeutic approaches for cancer.

## 1. Introduction

The human gastrointestinal microbiota comprises the bacteria, archaea and eukarya species colonizing the human gut. Moreover, the total amount of genetic material exceeds that of the genetic material of the human host [1]. There are more than 1000 presently known species, distributed mainly among the phyla *Bacteroidetes* and *Firmicutes* (around 90%) and the rest among *Proteobacteria, Verrucomicrobia* and *Actinobacteria* [2]. However, the human gut microbiota demonstrates high geographical, interpersonal and age variability [1,2]. Moreover, its genetic and metabolomic abundance comprises a more extensive set of genes than the entire human genome [3]. The latter justifies the numerous metabolic and immunological interactions between the gut microbiota and the host, with variations in plasma metabolites and the host immune system activity correlating to the gut microbiota signature [4,5,6]. These host–microbiota interactions have local and systemic impacts, including on the pathological processes of carcinogenesis. 

It has been demonstrated that carcinogenesis could be promoted or suppressed by the interaction between the microbiome and the host. For example, a considerable amount of evidence for the role of the gut microbiota and intestinal dysbiosis in the development of colorectal cancer exists [7,8]. There is also mounting evidence for the significance of gut microbiota for other non-intestinal locations of carcinogenesis, such as breast and lung cancer [9,10]. 

While the potential gut microbiota signatures are endless and numerous correlations can be made, in our opinion, it is convenient to approach the complex interactions between gut microbiota and cancer from the viewpoint of the major oncogenic and tumor-suppressive pathways involved. Therefore, this paper aims to review the most up-to-date information on the complex tumor-promoting and tumor-suppressing interplay between gut microbiota and cancer. Furthermore, we aimed to provide evidence that interactions between the host and the gut microbiota might be a significant exogenic factor for cancer predisposition.

Our search strategy was as follows. We conducted a modified form of a biomedical narrative review according to recent recommendations [11]. We performed a search through scientific databases Medline (PubMed) and Scopus. Both MeSH and relevant free-text terms were used: (“microbiome” OR “gut microbiota”) AND (“oncogenesis” OR “carcinogenesis”). Additionally, we searched for (“microbiome” OR “gut microbiota”) AND “colorectal cancer”, “interleukin pathways”, “NF-kB”, “Th17 cells”, “tumor suppression” and “HDACs”. Our search was confined to articles published up to January 2023. Secondly, relevant data were also derived from sources using the search engine Google Scholar. Finally, the references of retrieved publications were further hand-searched for supplements.

## 2. Local Oncogenic Effects of Gut Microbiota

The gut microbiota can promote local colonic oncogenesis through the production of cancerogenic metabolites, oncogenic exotoxins and the axis of chronic inflammation, including biofilm production, pathogenic adhesins stimulation and local immune system mediation [12]. 

For example, the enterotoxigenic *Bacteroides fragilis* strains (EBFT) produce one of the three isotypes of a 20 kDa zinc metalloprotease (BFT-1, BFT-2 and BFT-3). BFT modify the permeability of the colonic epithelium, inducing colitis, and promote cell proliferation through the NF-kB and the MAPK pathways [13]. In addition, it was revealed in a mouse model that BFT enhance the oncogenesis of colorectal cancer (CRC) through the Th17 pathway, and EBFT has been linked to an increased risk of CRC in human patients [12,13]. Recently, our team demonstrated that upregulated IL-6 is crucial for both inflammatory bowel diseases and CRC development, whereas Th17/T regulatory (Treg) cells and related genes are activated primarily in CRC [14]. It has been suggested that chronic inflammation mediated by Th17 cells and related cytokines is associated with an increased risk of malignant transformation.

Another prominent group of microbiota toxins interferes with the DNA and cell cycle of the intestinal epithelium. Colibactin is a byproduct of the *pks* genetic island in some *Enterobacteriaceae*, particularly *Escherichia coli*. It is an unstable compound, which, when introduced directly on the mucosal surfaces or intracellularly, causes DNA alkylation, interstrand crosslinking and double-strand breaks [15,16]. Then, DNA damage translates into mutagenesis and carcinogenesis potential, which has been demonstrated on epithelial cell lines and linked to colon cancer risk [17,18]. 

A Swedish study discovered colibactin-producing bacteria in 56% of the CRC samples versus 19% of control samples [19]. The cytolethal distending toxin (cdt) is another potent bacterial toxin isolated initially from *E. coli* and consequently described in *Shigella, Campylobacter* and other Gram-negative bacteria. It can be coded both on the bacterial chromosome, but also on plasmid vectors. Cdt is a virulence factor that enhances the invasive properties of its carriers through damage to the epithelial layer of the intestinal mucosa. Cdt also extensively suppresses lymphocytes and macrophages [20,21,22]. The mechanism of action of cdt relies on its structural similarity to the human DNAse I protein family that leads to cell cycle damage through the infliction of double-strand breaks. The described processes are often observed in mutagenesis and carcinogenesis [23]. It has been demonstrated in animal models that mice bearing cdt-positive *Campylobacter jejuni* are more likely to develop CRC and larger tumors [24]. 

The typhoid toxin (TT) found in *Salmonella* also induces double strand breaks in a similar manner, as the active part of TT actually comprises the cdt B subunit. However, TT is unique in being directly linked to an increased risk of developing CRC in humans. A recent Dutch study demonstrated a statistically significant increase in standardized incidence risk for CRC development after salmonellosis infection. The overall risk increased by 1.54 times for the age group above 20 and below 60 years, while for the age groups 20–39 and 40–49, the increases were 2.55 and 1.62 times, respectively [25,26]. 

Gut microbiota can also promote oncogenesis in a non-DNA-related way; intestinal commensals can promote inflammation-related epithelial cell proliferation in an IL-6 STAT3-dependent way, in IL-17C dependent way or through the PI3K-Akt-axis [27]. These proinflammatory pathways relate to specific virulence factors enhancing biofilm production, specific adhesins and immune cell recruitment in the tumor stroma. For example, *Fusobacterium nucleatum* was related to IL-17A-related inflammation in human CRC, and *Peptostreptococcus anaerobius* was described to induce a proinflammatory response in the tumor stroma, contributing to tumor progression via the recruitment of tumor-infiltrating immune cells [27,28].

The above mechanisms are further implied by novel sequencing techniques suggesting there are specific gut microbiota signatures in CRC patients. A few studies have demonstrated the loss of alpha diversity of the gut microbiota of CRC patients [29,30,31]. A large meta-analysis of 386 samples from CRC patients and 392 tumor-free controls revealed heterogenic data on the alpha diversity but demonstrated the prevalence of 29 bacterial species in the CRC samples, among which were the already discussed *Fusobacterium* and *Peptostreptococcus* [32]. The same study also demonstrated significant enrichment in the pks gene already discussed above and also of the fadA gene (and adhesin and virulence factor of *F. nucleatum*). Another large study demonstrated a correlation between a western style diet and high levels of pks positive *E. coli* and CRC [33]. Additionally, while some of the largest of these studies demonstrate significant correlations in samples from cancer patients versus healthy controls, further prospective long-term studies will be needed to further elaborate on the role of metagenomic profiles and their contribution to the development of CRC and to exclude the potential reciprocal relationship where CRC impacts the local flora.

## 3. Systemic Oncogenic Effects Related to Gut Microbiota

While the local effects of gut microbiota on the intestinal epithelium, causing local inflammation and induction of IL-17 mediated stimulation of the NF-kB pathway towards enhanced proliferation, are well established, it is interesting to know if these pathways can have a more systemic impact other than on intestinal epithelium tissues [34]. For example, increased IL-17 in the bone marrow microenvironment can promote the nuclear factor kappa-light-chain-enhancer of activated B cells (NF-kB), which in turn enhances disease activity in particular bone marrow cancers, such as multiple myeloma (MM) [34]. 

Short-chain fatty acids (SCFAs) are gut microbiota products that may have anti-inflammatory properties, locally and systemically. Dysbiosis shifting the intestinal microbiota balance towards bacterial species producing less SCFA, such as *Klebsiella* spp., could change the balance towards more prominent NF-kB activation. These observations were described in MM patients [34]. 

*Prevotella heparinolytica*, as a distinct member of the gut microbiota, is also related to the stimulation of the Th17 response, promoting NF-kB activation in the bone marrow and enhancing progression from monoclonal gammopathy of undetermined significance (MGUS) to MM [35,36]. 

In line with this, the NF-kB pathway is involved in many systemic hematological malignancies such as chronic lymphocytic leukemia (CLL), mucous membrane-associated lymphoma (MALT), diffuse large B-cell lymphoma (DLBCL) and acute lymphoblastic leukemia (ALL) [37]. Additionally, while the direct prognostic link between human hematological malignancies and specific gut microbiota signatures has yet to be demonstrated, a few studies have shown decreased gut microbiome diversity in patients with hematological malignancies [35]. 

A small American study of 51 pediatric and adolescent subjects (ALL patients and their healthy siblings) revealed a reduced microbial diversity of the gut flora in ALL patients compared to the healthy controls [38]. In addition, a Chinese group compared 25 DLBCL patients at the time of diagnosis to 26 healthy controls, demonstrating an increased abundance of *Escherichia-Shigella, Enterococcus, Veillonella* and *Prevotella* at the genus level in the lymphoma patients and a different beta-diversity pattern between DLBCL patients and healthy controls [39].

The spreading of tumors has also been related to the gut microbiota. Several mechanisms and strategies are employed by microbiota to regulate cancer progression and metastasizing.

Chronic exposure to bacterial metabolites causes chronic inflammation if the epithelium’s barrier function is compromised or if the tumor or metastases are persistently colonized with bacteria. Inflammation can then promote tumor initiation, development, progression and metastasis via pathways linked to cancer stem cell compartment control, immunosuppression, mutation induction, tumor microenvironment modulation and stromal cell modulation [40]. One of the most prominent mechanisms is bacteria’s ability to activate innate and adaptive immune cells within the tumor environment via Toll-like receptors, NOD-like receptors and other pattern recognition receptors to stimulate the production of pro-inflammatory cytokines, which not only regulate immunity and the microenvironment, but also serve as tissue-protective and repair responses in epithelial and cancer cells [41].

Additionally, in response to microbe-driven inflammation, cell plasticity causes cells to move from one stage of differentiation to another, promoting epithelial-mesenchymal transition and metastasis. Microbiota and microbial products can govern metastatic survival locally (e.g., in the liver, which is often rich in microbe-derived signals) or systemically, as in the case of metastasis in the lungs. By activating inflammation-driven NF-kB signaling, lipopolysaccharide (LPS) was demonstrated to enhance the survival and expansion of cancer cells metastasizing to the lung. LPS also increased metastatic dissemination by enhancing cancer cell integrin-mediated adherence to endothelial cells in vessels [42]. 

The interaction between the host`s immune system, microbiome and cancer cells is presented in Figure 1.

## 4. Tumor Suppressor Effects Related to Gut Microbiota

Significant amount of research has recently been directed toward elucidating how commensal bacteria impact antitumor immunity. Several hypotheses speculate that the gut microbiome has the potential to enhance the host’s immune responses against tumor cells, which are covered in detail in the following paragraphs.

The immune system evolved as an evolutionary protective mechanism to recognize and remove neoplastic cells, preventing cancer from spreading to other organs. Tumor immune surveillance involves a variety of immunological effector pathways, including innate and adaptive mechanisms [43]. Long-term chronic activation of immune cells by tumor antigens and uncontrolled inflammation associated with carcinogenesis, on the other hand, might eventually weaken antitumor immunity and accelerate tumor growth. As a result, tumor cells’ capacity to avoid and inhibit antitumor immunity is a characteristic of malignancy.

Furthermore, tumor cells avoid detection by suppressing antigen-presenting machinery and interferon (IFN) signaling pathways. Tumors also create an immunosuppressive microenvironment by recruiting myeloid-derived suppressor cells (MDSCs) and Tregs and producing pro-tumor and anti-inflammatory chemicals such as TGF-β, IL-10 and IDO [43]. 

However, the processes by which the gut microbiota impacts the efficacy of immunotherapy are currently being studied. Preclinical and preliminary clinical evidence from various gut microbiota modulation trials, including fecal transplantation, probiotics, consortia and nutrition, show that favorable microbiota modification is linked to enhanced intratumoral infiltration of CD8+ effector T cells. Furthermore, this infiltration of CD8+ T cells is frequently accompanied by the increased intratumoral activity of Th1 cells and dendritic cells and a decreased density of immunosuppressive cells [44].

One such proposed mechanism relies on the argument that microbiota-related antigens stimulate T cell reactivity against structurally similar tumor antigens based on the principle of cross-reactivity. Some evidence corroborating this hypothesis came from a study on long-term pancreatic cancer survivors [45]. 

By methods of whole-exome sequencing and in silico neoantigen prediction, the authors identified neoantigen-specific T cells in the peripheral blood and tumor-infiltrating lymphocytes which harbored specific reactivity to both tumor and bacterial epitopes [46]. This finding is part of a broader discussion of whether antigen mimicry between certain bacterial strains and tumor cells could lead to better anticancer immune surveillance. 

CD4+ and CD8+ T cells are primarily responsible for tumor growth and cancer progression. As a result, the effectiveness of some immunotherapy, such as immune checkpoint inhibitors (ICIs), primarily depends on T cells [43]. Notably, the gut microbiota is a tumor-extrinsic component that can influence the effectiveness of cancer immunotherapy using ICIs by modulating antitumor defensive systems, as we also demonstrated recently [47].

As discussed above, SCFAs, aryl hydrocarbon receptors (AhR) and the Toll-like and nod-like receptor (TLR and NLR) ligands are all microbiome-derived compounds that can regulate the immune system locally and systemically. Through dietary fiber fermentation, anerobic commensal bacteria of the *Bacteroidetes* and *Firmicutes* phyla create SCFAs such as acetate, propionate and butyrate. Butyrate and propionate, generated by fiber-fermenting commensal microorganisms, are connected to the formation of Foxp3+ Tregs by blocking histone deacetylases (HDAC) within the cell [48]. Although the in vivo instability of induced Tregs (iTregs) has been demonstrated previously, in the study by Souza Vieira et al., T cells may have lost FOXP3 expression throughout the sensitization stage and therefore did not exert a significant effect in their research.

Furthermore, the gut microbiome may stimulate adaptive immune responses by targeting tumor cells. For example, *Bifidobacterium* spp. stimulates tumor-specific T cells, boosts CD8+ T cell accumulation in melanoma and bladder cancers and increases IFN production. This may limit cancer cell proliferation by downregulating the NF-kB signaling pathway [49,50]. Furthermore, the *Bifidobacterium* strain improves antitumor immune responses in mice with colon cancer by raising CD4+ and CD8+ T cells and NK cells and the CD4+/Treg, CD8+/Treg and effector CD8+/Treg ratios [51]. Furthermore, the gut microbiome influences colitis-associated carcinogenesis by modulating the number of IFN-producing CD8+ T cells [52].

The *Prevotellaceae* and *Anaeroplasmataceae* families are predictive of high and low tumor burdens of colon cancer, respectively [52]. According to Li Y et al., bacterial strains, particularly *Bacteroides* and *Lactobacillus* spp., are related to better antitumor immunity, increased tumor infiltration by tumor-specific CD45+, CD4+, and CD8+ T cells and increased IFNg, TNFa and IL-2 production. This, in turn, may limit melanoma progression in Rnf5/ mice [53]. In general, the gut microbiota and its metabolites regulate antitumor immunity by many pathways, including Th1 and Th17 cell growth, production of pro-inflammatory cytokines and activation of MDSCs and NK cells [53].

Microbiota-derived epitopes may stimulate antigen-presenting cells for more T cell growth and cytokine production, thus boosting the systemic response to cancer immunotherapy [43]. Given the importance of gut microbiota in balancing antitumor vs. pro-tumor immune responses, developing microbiome screening and treatment techniques to help tilt the balance in favor of antitumor immunity is critical. Given the harmful effects of antibiotics on several essential components of innate and adaptive immunity, it is crucial to avoid broad-spectrum antibiotics as much as possible in cancer patients undergoing ICI treatment. More clinical research is needed to establish the potential efficacy of microbiome-supportive treatments such as fecal microbiota transplantation (FMT) from healthy donors in cancer patients who require antibiotic therapy [43].

However, while the cross-reactivity hypothesis sounds plausible, there are still some controversies surrounding identifying the exact epitope structure that drives T cell cross-reactions. Additionally, microbiome could probably do much more than stimulate T cell cross-reactivity against structurally similar epitopes. For example, recent research demonstrated that among the T cell repertoire, there are subpopulations of memory cells that possess specificity to neoantigens with entirely ‘unknown’ structures. The latter are usually antigens to which the immune system has not been previously exposed, and there is no antigen mimicry [46]. 

Memory T cells with specific reactivity to human immunodeficiency virus (HIV) were identified in blood samples from HIV-seronegative adults, but such cellular clones could not be detected at birth. Moreover, these memory immune cells are activated when treated with specific microbiome-produced peptides [46]. 

All this suggests that commensal gut bacteria might play a role in generating, sustaining and stimulating memory T cells against unencountered yet viral epitopes. In addition, one could argue that a similar response could be induced against neoepitopes emerging from tumor initiation and subsequent cancer progression.

Apart from the direct effects on the adaptive immune system, the microbiome can stimulate an antitumor immune response by rendering pattern recognition receptors on cells belonging to the innate immunity, e.g., dendritic cells (DC). In vitro experiments demonstrated that under the influence of bacterial strains such as *Lactobacillus*, DCs become activated, undergo maturation and begin producing cytokines (IL-12, IL18 and IFN-γ) which trigger cellular cytotoxicity [54]. 

*Lactobacillus* species differentially activate Toll-like receptors and downstream signals in dendritic cells. Some probiotic strains such as *Lactobacillus casei* BL23 could exert antitumor properties via the stimulation of the IL-2 signaling pathway. This was well presented in a study on HPV-induced cancers in mice [55]. It was first demonstrated in vitro how this bacterial species could stimulate bone marrow DC to produce IL-2. 

A recent study demonstrated the tumor suppressing properties of *Lactobacillus gallinarum* that were realized through the secretion of protective metabolites enhancing CRC cells apoptosis in mice models. Metabolomic studies and mass spectrometry revealed an increase in indole-3-lactic acid [56]. 

Thereafter, an in vivo analysis revealed an inverse correlation between IL-2 levels and tumor size, i.e., mice with higher levels of IL-2 had a much slower tumor growth. In addition to promoting IL-2-mediated antitumor activity, *Lactobacillus casei* BL23 was able to recruit natural killer (NK) cells with high cytotoxic potential against cancer. Other commensal species, such as *Bacillus mesentericus, Clostridium butyricum* and *Enterococcus faecalis,* can also lead to DC activation and enhanced antigen presentation of tumor antigens [57]. 

Furthermore, it was reported that when applied as a probiotic combination, these bacteria could skew CD4 differentiation towards the Th1 phenotype and stimulate peripheral blood mononuclear cells (PBMC) and DC. Another interesting fact is that the combination of microorganisms could induce immune changes on a grander scale than each species alone. This raises a question of potential synergism between different bacteria constituting normal gut flora.

An overview of the anticancer and tumor-promoting mechanisms of gut microbiota is presented in Figure 2.

## 5. Role of the Microbiota-Produced Metabolites

Commensal bacteria provide neoantigens to the host immune system and produce a wide range of metabolites, some of which have notable tumor-suppressing features. SCFAs are among the best-studied byproducts of the gut microbiome, with butyrate and propionate most famous for their antitumor properties. Butyrate is produced during the fermentation of fiber-rich food in the colon [58]. It plays an essential role in mitochondrial beta-oxidation. As tumor cells rely on alternative pathways for energy metabolism (aerobic glycolysis), they cannot utilize butyrate efficiently, and it accumulates in their nuclei. This leads to histone deacetylase (HDAC) inhibition and, ultimately, to tumor cell apoptosis [58]. 

In vivo experiments have demonstrated that when tumor-bearing mice are colonized with butyrate-producing bacteria, such as *Ruminococcus, Clostridium*, *Eubacterium* and *Coprococcus,* their tumors exhibit diminished tumor growth compared to their counterparts whose microbiome is incapable of producing butyrate [58]. Butyrate and propionate can also exert a local antitumor effect on intestinal mucosa by suppressing inflammation predisposed to carcinogenesis. In addition, both metabolites are known to interact with G-protein coupled receptors (GPCR) such as GPR109A and GRP43 [59,60]. 

After binding GPR109A, propionate induces inflammatory cytokine secretion from colonic macrophages and dendritic cells. This leads to T cell differentiation towards the immunosuppressive T regulatory phenotype. Butyrate also has an affinity to GPR109A, which is how this metabolite stimulates the production of IL-18, a cytokine with a defined role in preventing the development of colon tumors [61]. On the other hand, butyrate suppresses IL-18 production by suppressing caspase 1 or NLRP3 inflammasome [62].

Moreover, butyrate has remarkable anticancer properties due to its potential to skew a negative impact on the cell development of malignant colonocytes while being the primary energy source for normal colonocytes. It was shown that butyrate suppresses glucose metabolism of colorectal cancer cells via the GPR109a-AKT signaling pathway, thus enhancing chemotherapy [63].

Data from in vivo models of GPR109A-deficient mice support the antitumor properties of SCFAs. These animals showed an increased propensity to form colon tumors, suggesting that interactions between SCFA and GPCR might be responsible for the tumor-protective effect of a healthy diet and normal gut flora [64]. There is also speculation that the balance between pro- and anti-inflammatory regulation depends on the concentration of SCFAs, where lower levels of butyrate could induce T reg differentiation and higher concentrations could skew T cells towards the inflammatory Th1 phenotype [64]. 

Besides the production of metabolites, the gut microbiome is also involved in metabolizing endogenous substances such as bile acids. Under the influence of intestinal flora, primary bile acids are converted to secondary bile acids, and both (primary and secondary bile acids) can exert immunoregulatory effects, mostly in the liver, where they enter via the enterohepatic circulation [65]. 

Recent data suggest that by the controlling bile acid metabolism, the gut microbiome could either augment or impair local hepatic antitumor immunity [66]. Authors have reported that manipulating the gut bacterial composition in mouse models could increase the accumulation of CXCR6+ NK cells in the liver and enhance interferon-y production. Moreover, these NK cells showed selective antitumor activity against liver malignancies [66]. A proposed mechanism behind CXCR6+ NK accumulation in the liver is based on the microbiome-controlled conversion of primary bile acids into secondary ones. It was demonstrated that when mice were treated with antibiotics against Gram-positive bacteria, this impaired bile acid conversion and increased primary bile acids. The latter led to increased CXCL16 expression from sinusoidal endothelial cells, which bonded to CXCR6 on CXCR6 + NK cells and enhanced their accumulation in the liver [66].

Other metabolites, such as trimethylamine-N-oxide (TMAO), hydrogen sulfide (H_2_S), N-nitroso compounds (NOCs), DCA, deoxycholic acids, HCAs, heterocyclic amines, etc., are critical signaling molecules that mediate the cross-talk between the host and the microbiota and play a significant role in colorectal carcinogenesis [67].

Furthermore, these metabolites are related to the etiology and severity of CRC. It was shown that some of these metabolites could act directly on the integrity of the mucosal barrier and immune responses in the gut, triggering the release of proinflammatory cytokines (i.e., TNFa and IL-17) [67]. This usually correlates with immune escape and the immunosuppression state that promote tumorigenesis. Additionally, the mentioned metabolites may damage DNA via several pathways. For example, DCAs promote tumor formation through the RAS-ERK1/2 signaling pathway, the Wnt/b-catenin signaling pathway and the PKC/p38 MAPK signaling pathway. Trimethylamine (TMA), produced from choline, carnitine and phosphatidylcholine from food processing by microbiota and then transported and oxidized to TMAO in the liver, increases the risk of developing CRC. Experimental models showed that hydrogen sulfide damages DNA through free radical oxygen and genotoxicity. NOCs also lead to DNA damage and NOC-induced DNA adducts by different mechanisms: inducing mutations in the K-ras gene (G→A) and oxidative stress. Experimental studies showed that HCA is also associated with DNA damage and the formation of DNA adducts. Interestingly, lactate metabolites also promote CRC cell proliferation, invasion and migration by providing environmental conditions and stimulating the glycolytic metabolism and angiogenesis [67].

The microbiota-derived metabolites are presented in Table 1.

It is essential to mention that these metabolites usually act together. For example, TMAO participates in NOC formation, leading to DNA damage and epigenetic changes [67].

In numerous studies, however, alteration of the gut microbiota has been linked to gastrointestinal disorders such as CRC, although cancer is considered a multifactorial and multistage disease. However, the function of microorganisms in the initiation and progression of colorectal cancer has become apparent. Convincing models have been presented to depict the complex and dynamic mechanisms and shifts in carcinogenesis [12,100]. 

Gut microbiota metabolites have also been assessed in human serum through untargeted and targeted metabolic approaches. A recent comprehensive metabolomic study among 44 healthy adenoma and CRC subjects demonstrated the distinct metabolic signature of patients with adenoma and CRC, correlated to the distinct increased bacterial species in adenoma and CRC subjects [101].

## 6. Histone Deacetylases Pathways and Tumor Cell Apoptosis

Microbiota-related epigenetic changes include DNA and histone modifications (i.e., acetylation and methylation) and non-coding RNAs (i.e., microRNAs and miR). Epigenetic regulation by the gut microbiota has been intensively studied in recent years, with many mechanisms and clinical implications demonstrated [102,103,104].

It was demonstrated that the gut epithelium secretes a range of miRNAs which can penetrate bacteria and impact their transcription and change the microbial community’s structure and diversity. Similarly, the gut microbiota produces several metabolites (i.e., butyrate and bile acids) which can modulate the human metabolism, including BMI, insulin secretion and fat synthesis. Dietary nutrient intake (e.g., folic acid, methionine and vitamin B12) supplies methyl donors that change host DNA methylation, which may modulate the inflammatory state of the intestine. Specific microbial metabolites can alter DNA methylation, histone acetylation and miRNAs, impairing intestinal homeostasis and suppressing beneficial microbiota while boosting the richness of harmful bacteria and promoting colorectal cancer [104,105]. 

We summarize the cross-talk between microbiota and epigenetic modifications in Figure 3.

The role of epigenetic modifications by histone-modifying enzymes in terms of cellular transformations is also an area of intensive research. Histone acetylation is a chromatin modification generally associated with opening the chromatin and allowing access to transcription factors. Acetylation happens on lysine residues by histone acetylases (HATs) and is reversed by histone deacetylases (HDACs) which have been found to play a particular role in aging and carcinogenesis. HATs and HDACs themselves are subject to regulation by post-translational modifications, protein–protein interactions, availability of cofactors and various signaling pathways [106]. 

The interaction between histone acetylation status and the gut microbiome is now recognized, although this was observed decades ago [102,107]. In the growing amount of possible molecular biomarkers relating gut microbes to health and disease, HDACs are attracting more attention due to the development of HDAC inhibitors as potential antitumor therapeutic agents [107]. 

HDACs may be involved in CRC onset and development in at least three directions. 

Firstly, SCFAs, the primary metabolites produced by the gut microbiota, may act alone as HDAC inhibitors and thus promote a more transcriptionally active state of chromatin [108]. In addition, butyrate, propionate and acetate produced by the microbiota have been shown to act as such inhibitors and also increase the cytokine production of Th1 and CD4+ T cells, which could influence apoptosis and cell cycle arrest of cancer cells [109]. 

Another significant histone modification in the intestinal epithelium, crotonylation, is also stabilized by butyrate by inhibiting class I HDACs, which are related to cancer pathways [110]. Furthermore, the activity of HDAC3 that is highly expressed in the intestinal epithelium is also affected by SCFAs and seems to integrate multiple microbiota-derived signals [111,112]. Similar data are available for HDAC1 and HDAC2 [113,114,115]. Furthermore, HDACs may also control cell proliferation and apoptosis via modulation of the acetylation status of non-histone proteins such as p53 and tubulin [116].

## 7. Cancer Immunotherapy and Microbiome

Recent preclinical studies using cell culture and animal models, human clinical studies, and meta-analyses of clinical studies have revealed that the gut microbiota alters the host’s response to various anticancer drugs, with immunomodulation emerging as one of the central mechanisms facilitating these differential responses. Dysbiosis is not only a result, but also a cause, of varied reactions to treatment. Increased intestinal diversity, for example, was predictive of a lower mortality in patients receiving allogeneic hematopoietic stem cell transplantation (allo-HSCT) for treating hematological malignancies [117]. 

The fact that immunological modulation caused by increased microbial diversity determines the severity of graft vs. host illness is an essential factor for patients starting allo-HSCT. Furthermore, compositional changes caused by therapy might also be involved [118,119]. For example, checkpoint inhibitors, such as other cancer medicines, have significant inter-individual heterogeneity in patient responses [47]. 

A rising amount of research has already offered considerable insight into the effect of the gut microbiota on the xenobiotic metabolism, which might significantly impact future disease treatment. For example, gut microbial xenobiotic metabolites have altered bioavailability, bioactivity and toxicity [120,121]. In addition, they can interfere with the functions of human xenobiotic-metabolizing enzymes to influence the fate of other exogenic molecules.

However, bacteria may become pharmacological targets in the not-too-distant future. Microbial drug targets may also reduce the negative effects of many chemotherapeutics on the GI tract (i.e., adverse effects, such as those caused by irinotecan (camptothecin)). For example, the microbiota can reactivate the active form of the chemotherapeutic agent (SN38) in the GI lumen, which is then glucuronidated in the liver to generate the inactive SN38-G, which is eliminated via the GI system. Then, increased SN38 levels in the colon produce severe and possibly fatal diarrhea, necessitating dosage reduction and frequent dose adjustments [122]. In addition, incorporating fiber-rich, prebiotic foods, restricting red meat intake and lowering obesity rates could assist in reducing the global tumor burden in the long run by the beneficial effects caused by these actions, such as the production of SCFAs during the bacterial fermentation of plant-based fibers, which provide broad protection against the development of cancer [122].

Since dysbiosis appears to be a carcinogenesis precursor, it not only predates illness onset but also spreads during tumor growth. Therefore, maintaining eubiotics, or an optimum microbiota composition, is critical for avoiding disease-causing events. As a result, new, specialized, narrow-range antibiotics that preferentially target infections or pathobionts while maintaining eubiosis are needed [118,123,124,125,126]. 

Speaking of microbiota, precision medicine that offers medical therapies tailored to each patient’s genetic makeup and variances in lifestyle and environment also pays attention to microbiome interventions [127,128]. Given the wide variety of impacts that the microbiota has on human health, changes in patient composition should be considered when determining who will benefit from a given treatment technique. For example, as previously stated, the presence or absence of particular bacterial community members, or even their metabolites, can affect the occurrence, severity and therapy of cancer and may serve as prognostic biomarkers. Furthermore, greater access to centralized, cloud-based repositories for whole-genome and transcriptome sequencing information would aid computer scientists’ data mining methodologies. Combining pharmacogenomics data with unique microbial organisms or their particular metabolites will most likely allow for accurate dosage, symptom control and enhanced treatment responses in the future [129]. 

Furthermore, because antibiotics might alter the makeup of the gut microbiota, interfering with the impact of immunotherapy, the link between antibiotic-associated dysbiosis and immunotherapy is a hot issue. For example, Elkrief et al. discovered that using antibiotics before immunotherapy, such as anti-PD-1/PD-L1 and anti-CTLA-4, was an independent risk factor for lower progression-free survival (hazard ratio = 0.32, 95% confidence range = 0.13–0.83, *p* = 0.02) [128]. Furthermore, patients who received antibiotics before immunotherapy had a lower chance of responding effectively and a better prognosis. 

Aside from using the gut microbiota to predict immunotherapy success, several clinical trials have focused on adjusting the gut microbiota composition to overcome anti-PD-1/PD-L1 resistance, including using FMT [129]. Even though numerous preclinical studies have shown that the gut microbiota regulates the host systemic immune response, modulates immunotherapy efficacy and influences treatment-induced adverse effects, the regulatory function of certain commensal bacteria requires further investigation, particularly for extrapolation from the mouse model to humans. The findings of these ongoing investigations may give more consistent data to demonstrate the viability of improving immunotherapy efficacy by modifying the gut microbiota makeup. It is worth noting, however, that the original gut mucosa commensals interfere with the colonization of supplementary probiotics. In addition, resistance to probiotic colonization varies amongst populations and may be impacted by baseline commensal status. As a result, the patient’s commensal background should be considered when modifying gut microbiota through therapies such as fecal transplantation [129]. 

In the case of CRC, several approaches have been investigated to target and modulate the gut microbiota composition, including both microbial physiology and/or their metabolites that cause or contribute to CRC directly or indirectly, for example, dietary interventions, antibiotic treatments, probiotics, prebiotics and postbiotics, as well as FMT. In addition, several experimental investigations have advanced our understanding of the function of gut biomodulators and microbe-based therapy as anticancer agents. Still, a practical clinical application in CRC prevention and management remains mainly missing. Finally, studies are being conducted to determine the effectiveness of tailored diets and biomodulators in restoring a eubiotic state to prevent and treat CRC [130,131], since the microbiota could facilitate cancer progression [132,133].

Our review has some limitations associated with the nature of the narrative review, and the data were not further analyzed statistically. However, we did our best to present the data comprehensively and discuss the studies critically. Furthermore, among the strengths of our review is that we gave the information logically by showing the data from animal studies and then described clinical trials and investigations that translated big data into practice. In addition, our review agglomerates gut microbial metabolites and signaling pathways of the related immune process, which has not been covered elsewhere.

## 8. Conclusions

In this review, we presented the available data and evidence supporting the idea that interaction between the host and the gut microbiota might be a significant exogenic factor for cancer predisposition. Various mechanisms by which bacteria could induce or prevent carcinogenesis are attaining more and more attention these days. The more we investigate, the more we learn about how diverse these mechanisms and pathways could be. We already know that the balance between commensal and pathogenic gut bacteria could affect intestinal mucosa permeability, local and systemic immunity and inflammation. 

In addition, bacterial metabolites could stimulate either pro- or anti-tumor processes. Finally, it is beyond doubt that the cross-talk between microbiota and the host cells in terms of epigenetic modifications can regulate gene expression patterns, which may further influence cell fate in beneficial or adverse directions. However, the exact mechanisms behind these interactions are elusive. Therefore, to better understand them and possibly discover new therapeutic approaches, there is a need for large-scale omics studies.

## Figures and Tables

**Figure 1 ijms-24-05978-f001:**
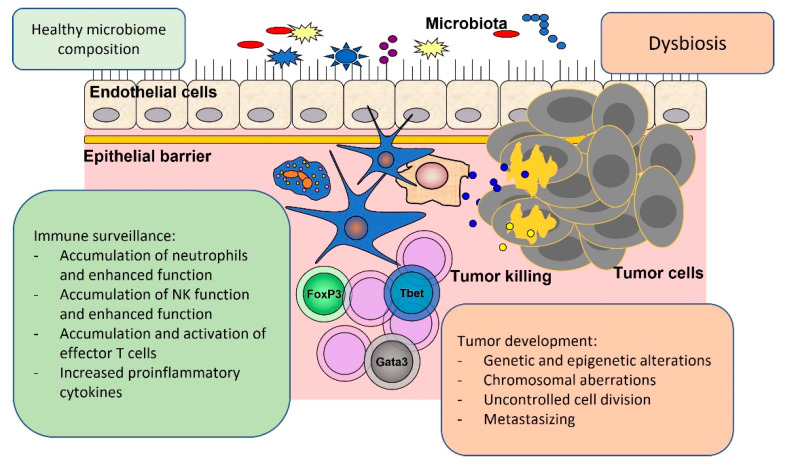
Immune surveillance and tumor development are connected with the state of health of the microbiome composition or dysbiosis.

**Figure 2 ijms-24-05978-f002:**
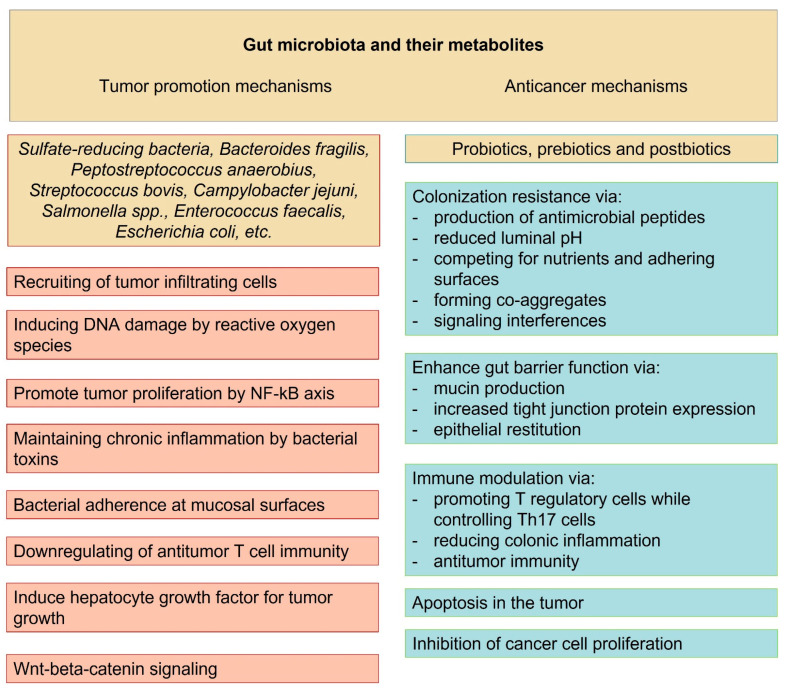
Tumor promoting and anticancer mechanisms and pathways of gut microbiota and their metabolites.

**Figure 3 ijms-24-05978-f003:**
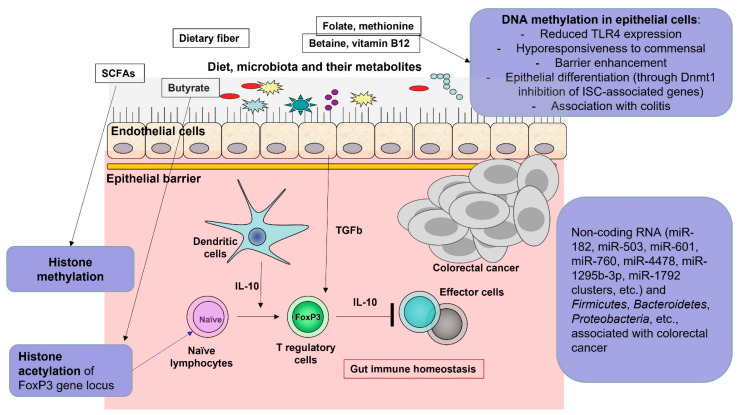
Cross-talk between microbiota and epigenetic modifications associated with gut homeostasis or disease (including cancer). Arrows represent influence or development, and an arrow followed by | represents suppression.

**Table 1 ijms-24-05978-t001:** Microbiota-derived metabolites, their primary sources and their effects on carcinogenesis.

Microbiota-Derived Metabolite	Source	Effects	References
SCFA	Anaerobes	Antitumor effect, reducing inflammation	[58,59,60,61,62,63,64]
Secondary bile acids	*B. fragilis*, *Bacteroides vulgatus, Clostridium perfringens, Eubacterium, Lactobacillus and Bifidobacterium*	Contribute to CRC progression	[65,66,68]
Indoles	Gram-positive anaerobe (i.e., *P. anaerobius*)	Tumor prevention	[69,70]
TMAO	Gram-negative, Gram-positive, anaerobe (i.e., *E. coli, Clostridium* and *Desulfovibrios*)	Positive association with CRC risk, new prognostic marker	[71,72]
H_2_S	Gram-negative anaerobes (i.e., F. *nucleatum* and *Desulfovibrios*)	Potential environmental risk factors for CRC	[73,74,75]
DCA	Gram-positive, Gram-negative anaerobes (i.e., *Desulfovibrios* and *Clostridium*)	Positive associations with colorectal adenomas and CRC; contributes to CRC development and carcinogenesis promotion	[76,77,78,79,80,81,82,83,84]
NOCs	Facultative and anaerobes	Positive association with CRC risk	[85,86,87,88,89]
HCAs	*Bacteroides, Lactobacilli*	Positive association with CRC risk; *Bacteroides* convert HCA to carcinogens and *Lactobacilli* reduce their mutagenic effect	[90,91,92]
Polyamines	Gram-negative anaerobes (i.e., *B. fragilis* and *F. nucleatum)*	Positive association with CRC	[93,94,95]
Ammonia	Gram-negative anaerobes, clostridia, enterobacteria and *Bacillus* spp. Gram-positive non-sporing anaerobes, streptococci	Contribute to CRC development and promote neoplastic transformation	[96,97]
Lactate	Lactobacillus, Leuconostoc, Pediococcus, Lactococcus and Streptococcus	Promote CRC	[98,99]

SCFAs—short-chain fatty acids; H_2_S—hydrogen sulfide; TMAO—trimethylamine-N-oxide; DCA—deoxycholic acids; HCAs—heterocyclic amines; NOCs—N-nitroso compounds; TMAO—trimethylamine-N-oxide; TMA—trimethylamine.

## Data Availability

Not applicable.

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
