# Peer review of "Gut Microbiome Composition and Its Metabolites Are a Key Regulating Factor for Malignant Transformation, Metastasis and Antitumor Immunity"

_ijms, 2023, doi:10.3390/ijms24065978_

Round 1
Reviewer 1 Report (Previous Reviewer 2)
The article by Stefan Lozenov et al. entitled "Gut microbiome composition and their metabolites are a key regulating factor for malignant transformation, metastasis, and anti-tumor immunity," still raises the following issue.
1. The title of the article is misleading since the authors did not provide adequate shreds of evidence to rationalize gut microbiota as a "key regulating factor" in malignant transformation and metastasis.
2. The aim of the study in the abstract and introduction was different and contradictory, which made readers confusion
3. In order to prove the hypothesis, crosstalk between microbiota and epigenetic modifications was not fully demonstrated. Need to elaborate on the literature using tables and figures
4. Section 2. Microbiota-induced oncogenesis. Local oncogenic effects of gut microbiota. Is it two titles?
5. Section 4. Tumor suppressor effects related to the gut microbiota. Anti-cancer immune response promotion, Again, two titles?
6. Section 4. Authors can provide detailed illustrations to substantiate the gut microbiome and their anticancer mechanisms
7. Authors must check carefully for grammatical, stylistic, and typographical errors.
8. There were many studies available related to this manuscript in Pub Med, thus no novelty. The manuscript was poorly written, and the title and hypothesis were controversial. The listed references are quite related to the topic, which is in PubMed: Front Immunol. 2021 Mar 26;12:612826. doi: 10.3389/fimmu.2021.612826; Int J Mol Sci. 2019 Oct 24;20(21):5295. doi: 10.3390/ijms20215295; CA Cancer J Clin. 2017 Jul 8;67(4):326-344. doi: 10.3322/caac.21398; Sci Transl Med. 2015 Jan 21;7(271):271ps1. doi: 10.1126/scitranslmed.3010473; Eur J Clin Microbiol Infect Dis. 2017; J Cancer Res Clin Oncol. 2019 Jan;145(1):49-63. doi: 10.1007/s00432-018-2816-0. May;36(5):757-769. doi: 10.1007/s10096-016-2881-8. Front Immunol. 2020 Nov 30;11:615056. doi: 10.3389/fimmu.2020.615056; Clin Transl Oncol. 2022 Nov 14. doi: 10.1007/s12094-022-02995-5. .
Author Response
The article by Stefan Lozenov et al. entitled "Gut microbiome composition and their metabolites are a key regulating factor for malignant transformation, metastasis, and antitumor immunity," still raises the following issues.
- The title of the article is misleading since the authors did not provide adequate shreds of evidence to rationalize gut microbiota as a "key regulating factor" in malignant transformation and metastasis.
- Thank you for the critical point. In our view, we have presented a comprehensive review demonstrating the complex interplay between gut microbiota and cancer. In our review, the bulk of evidence suggests a key role for gut microbiota in malignant transformation, malignant progression and antitumor immunity.
- If the word “key” is inappropriate, we can remove it, leaving “Gut microbiome composition and their metabolites are a regulating factor for malignant transformation, metastasis and antitumor immunity”
- However, our review is limited in volume by the article volume limitations, but within the scope of an article review, we consider it comprehensive and up-to-date. Furthermore, we previously have revised the title to make it more precise and not mislead the audience.
- The aim of the study in the abstract and introduction was different and contradictory, which made readers confusion
- Thank you for noticing this. We have revised the aim in the text to align entirely with the objective in the abstract.
- In order to prove the hypothesis, crosstalk between microbiota and epigenetic modifications was not fully demonstrated. Need to elaborate on the literature using tables and figures
- Thank you for the excellent suggestion. We agree that the topic of microbiota and epigenetics needs to be extended. So we added a figure and a summary of the findings while focusing mainly on the histone modifications.
- Section 2. Microbiota-induced oncogenesis. Local oncogenic effects of gut microbiota. Is it two titles?
- We have corrected and simplified section headings to avoid double headers.
- Section 4. Tumor suppressor effects related to the gut microbiota. Anti-cancer immune response promotion, Again, two titles?
- We have corrected and simplified section headings to avoid double headings.
- Section 4. Authors can provide detailed illustrations to substantiate the gut microbiome and their anticancer mechanisms
- Thank you for the suggestion. We crafted a figure to illustrate the gut microbiome, its anticancer, and tumor-promoting mechanisms.
- Authors must check carefully for grammatical, stylistic, and typographical errors.
- We agreed with the referee and revised the paper more profoundly for any remaining errors.
- There were many studies available related to this manuscript in Pub Med, thus no novelty. The manuscript was poorly written, and the title and hypothesis were controversial. The listed references are quite related to the topic, which is in PubMed: Front Immunol. 2021 Mar 26;12:612826. doi: 10.3389/fimmu.2021.612826; Int J Mol Sci. 2019 Oct 24;20(21):5295. doi: 10.3390/ijms20215295; CA Cancer J Clin. 2017 Jul 8;67(4):326-344. doi: 10.3322/caac.21398; Sci Transl Med. 2015 Jan 21;7(271):271ps1. doi: 10.1126/scitranslmed.3010473; Eur J Clin Microbiol Infect Dis. 2017; J Cancer Res Clin Oncol. 2019 Jan;145(1):49-63. doi: 10.1007/s00432-018-2816-0. May;36(5):757-769. doi: 10.1007/s10096-016-2881-8. Front Immunol. 2020 Nov 30;11:615056. doi: 10.3389/fimmu.2020.615056; Clin Transl Oncol. 2022 Nov 14. doi: 10.1007/s12094-022-02995-5. .
- Thank you for the critical note. We can understand your opinion of our paper as poorly written. We did our best to improve the manuscript by removing any flaws regarding the title, hypothesis, and aim of the review, adding more information on critical points, adding more illustrative materials and correcting grammar and style mistakes.
- Regarding novelty, on one side, we agree that there are some papers on the topic. Therefore, we thank the referee for the suggestions; we added four of the above-suggested by the referee papers, and the rest of them have already been cited in our paper. However, on the other side, although there are review articles on the gut microbiome and carcinogenesis, our review agglomerates gut-microbial metabolites and signaling pathways of the related immune process, which was not covered elsewhere, including in the cited papers.
- We can see that some of the points are severely critical. However, we hope that the reviewers` requirements will be satisfied since we addressed them appropriately.
Reviewer 2 Report (New Reviewer)
In the review article entitled “Gut microbiome composition and their metabolites are a key regulating factor for malignant transformation, metastasis and anti-tumor immunity, " Lozenov et. al gathered up-to-date information about the gut microbiome and its role in carcinogenesis. Updating our knowledge related to the human gut microbiome has become imminent more than ever due to their “friend or foe” role in human health. Recent studies suggested that the complex gut microbial flora harboring individual gut has been a contributing exogenic factor for individual health.
In this review authors put together the metabolites produced by gut microbes and their role in carcinogenesis and anti-tumor immunity. Though there are plenty of review articles on gut microbiome and carcinogenesis, this review agglomerates gut-microbial metabolites and signaling pathways of the related immune process.
Please rewrite the section headings for clarity.
Comments#
Line 55-56 – “Therefore, this paper aims to review the most up-to-date information on some of the most significant pathways related to carcinogenesis.” Is this review focused only on carcinogenesis-related pathways? I suppose this review is about “the role of gut microbiota in cancer and carcinogenesis-related pathways”.
Line 67 – What authors want to say here. The caption for this section is not clear. Either keep “Microbiota-induced oncogenesis OR Local oncogenic effects of gut microbiota”.
Line 72 & Line 84 &Line 99- Stick to the scientific nomenclature rule throughout the manuscript. Expand the genus name if it is mentioned for the first time.
Line 118 – What do authors want to say here “Metastasizing”. Probably add “and” in front of “Metastasizing” to make the heading meaningful
Line 177 – Again the heading is not clear. Seems the authors haven’t concluded the heading of each section after revision.
Lines 180-181 – Any references to support this statement.
Line 217- Expand TLR and NLR here
Line 222- Explain Foxp3+ regulatory T cells here. What is their role in immune response?
Line 225 – Change to “stimulate”
Line 359- What is “frood”? Mean food?
Line 448 & 477– Check the spelling of “patient”
Line 456 – Scientists’?
Author Response
In the review article entitled “Gut microbiome composition and their metabolites are a key regulating factor for malignant transformation, metastasis and antitumor immunity, “ Lozenov et. Al gathered up-to-date information about the gut microbiome and its role in carcinogenesis. Updating our knowledge related to the human gut microbiome has become imminent more than ever due to their “friend or foe” role in human health. Recent studies suggested that the complex gut microbial flora harboring individual gut has been a contributing exogenic factor for individual health.
- Thank you for your time reviewing our paper and the opportunity to improve it. We acknowledge that our paper might have some issues in conformity with the referees` comments. We have addressed them and revised the manuscript accordingly (with track changes).
In this review authors put together the metabolites produced by gut microbes and their role in carcinogenesis and antitumor immunity. Though there are plenty of review articles on gut microbiome and carcinogenesis, this review agglomerates gut-microbial metabolites and signaling pathways of the related immune process.
Please rewrite the section headings for clarity.
- Thank you for the valuable suggestion. We have corrected and simplified section headings to avoid double headings.
Comments#
Line 55-56 – “Therefore, this paper aims to review the most up-to-date information on some of the most significant pathways related to carcinogenesis.” Is this review focused only on carcinogenesis-related pathways? I suppose this review is about “the role of gut microbiota in cancer and carcinogenesis-related pathways”.
- Indeed, the review aims to present more wide data on the tumor promoting and tumor suppressing roles of gut microbiota, so the statement including only carcinogenesis-related pathways would probably be too narrow and inappropriate. We have revised that statement accordingly.
Line 67 – What authors want to say here. The caption for this section is not clear. Either keep “Microbiota-induced oncogenesis OR Local oncogenic effects of gut microbiota”.
- We have corrected and simplified section headings to avoid double headings.
Line 72 & Line 84 &Line 99- Stick to the scientific nomenclature rule throughout the manuscript. Expand the genus name if it is mentioned for the first time.
- This is a perfectly valid remark. Indeed during the article processing for submission, and several rearrangements of text, the italics and expanded names for microorganisms, appear to have been lost in some instances. We have corrected this.
Line 118 – What do authors want to say here “Metastasizing”. Probably add “and” in front of “Metastasizing” to make the heading meaningful
- We have corrected and simplified section headings to avoid double headings.
Line 177 – Again the heading is not clear. Seems the authors haven’t concluded the heading of each section after revision.
- We have corrected and simplified section headings to avoid double headings.
Lines 180-181 – Any references to support this statement.
- Thank you for the valuable comment. We revised the text to denote that these data and evidence will be covered in the following paragraphs in detail.
Line 217- Expand TLR and NLR here
- Thank you for noticing this issue. We have expanded the abbreviations.
Line 222- Explain Foxp3+ regulatory T cells here. What is their role in immune response?
- Thank you for the helpful note. We have expanded the information.
Line 225 – Change to “stimulate”
- Thank you for the recommendation. We changed the terms.
Line 359- What is “frood”? Mean food?
- Thank you for noticing this typo. We corrected it.
Line 448 & 477– Check the spelling of “patient”
- Thank you for pointing this out. Typo mistakes were corrected.
Line 456 – Scientists’?
- Thank you for noticing the typo. We corrected it.
Reviewer 3 Report (New Reviewer)
The authors wrote a quite interesting review on the microbiome, metabolisms, and cancer. This is generally of high interest. This is weak in discussion on some areas. Literature search is inadequate in those areas.
There are big research gaps. The authors fail to describe adequately. Foods and other factors are known to influence the gut microbiome, which in turn affect drug sensitivity. To address this, first, there are studies that examined foods / nutrients in relation to dysbiosis / pathogenic microbes. Second, there are studies that examined foods / nutrients in relation to risk of cancers such as colorectal cancer (CRC) and tumor progression. References in this area seem lacking. Third, there are studies that examined differences in stool microbiota between cancer patients and controls. Seminal meta-analysis studies in Nature Medicine 2019 (Wirbel et al.; Yachida et al.) are not mentioned in this area.
In any case, it seems that these study areas do not well connect with each other in many studies. There are indeed gaps. Molecular pathological epidemiology studies that examined foods/nutriest, CRC risks, and pathogenic microbes in tumors nicely can connect these gaps. See a recent study by Arima et al. Gastroenterology (2022). There are no other similar studies yet, but it will open new research approaches. However, this type of approach is rarely taken. There is a widely open opportunity that is currently missed. These facts should be discussed.
In these above contexts, the authors should discuss gaps and opportunities such as research on dietary / lifestyle factors, microbiome, and personalized molecular biomarkers, which is needed for further research, including research on drug resistance. The authors should discuss molecular pathological epidemiology research that can investigate diet, microbiota, and other factors in relation to molecular pathologies and clinical outcomes such as drug resistance. Molecular pathological epidemiology research can be a promising direction (eg, Gut 2022; etc.) and should be discussed in this paper.
Author Response
The authors wrote a quite interesting review on the microbiome, metabolisms, and cancer. This is generally of high interest. This is weak in discussion on some areas. Literature search is inadequate in those areas.
There are big research gaps. The authors fail to describe adequately. Foods and other factors are known to influence the gut microbiome, which in turn affect drug sensitivity. To address this, first, there are studies that examined foods / nutrients in relation to dysbiosis / pathogenic microbes. Second, there are studies that examined foods / nutrients in relation to risk of cancers such as colorectal cancer (CRC) and tumor progression. References in this area seem lacking. Third, there are studies that examined differences in stool microbiota between cancer patients and controls. Seminal meta-analysis studies in Nature Medicine 2019 (Wirbel et al.; Yachida et al.) are not mentioned in this area.
- We greatly appreciate this input. We have discussed in the section of local oncogenic effects of gut microbiota on CRC, the difference of the stool microbiota between cancer patients and controls. We have elaborated further on that and cited the seminal meta-analysis and additional studies as well. Our review is limited in its scope to the impact of gut microbiota on carcinogenesis, while we would consider the impact on drug sensitivity a separate topic.
In any case, it seems that these study areas do not well connect with each other in many studies. There are indeed gaps. Molecular pathological epidemiology studies that examined foods/nutriest, CRC risks, and pathogenic microbes in tumors nicely can connect these gaps. See a recent study by Arima et al. Gastroenterology (2022). There are no other similar studies yet, but it will open new research approaches. However, this type of approach is rarely taken. There is a widely open opportunity that is currently missed. These facts should be discussed.
- While our proposed review article aims to present a wider review of the effects of gut microbiota on many types of carcinogenesis both local and systemic, we indeed discuss the role of gut microbiota in CRC. It is a very interesting topic how food impacts gut microbiota, though we have tried to focus rather on how microbiota impacts cancer in our article entitled “Gut microbiome composition and their metabolites are a key regulating factor for malignant transformation, metastasis and antitumor immunity.” We have elaborated on the role of pks+ E. coli for CRC and we have reviewed the additional literature suggesting the higher incidence of pks+ E. coli in western diet. We have elaborated further in these interesting aspects as to improve further the quality of the article.
In these above contexts, the authors should discuss gaps and opportunities such as research on dietary / lifestyle factors, microbiome, and personalized molecular biomarkers, which is needed for further research, including research on drug resistance. The authors should discuss molecular pathological epidemiology research that can investigate diet, microbiota, and other factors in relation to molecular pathologies and clinical outcomes such as drug resistance. Molecular pathological epidemiology research can be a promising direction (eg, Gut 2022; etc.) and should be discussed in this paper.
- We have elaborated further on the role of gut microbiota for CRC which is indeed a section of our review for the role of gut microbiota in cancer. We have reviewed additional literature on the matter and elaborated further on the molecular markers and metabolome.
Round 2
Reviewer 1 Report (Previous Reviewer 2)
The article by Stefan Lozenov et al. entitled "Gut microbiome composition and their metabolites are a key regulating factor for malignant transformation, metastasis, and antitumor immunity," is quite interesting. However, the following issues need to be addressed.
1. The paper has been poorly designed, and information was not gathered and organized in a meaningful manner; therefore, the present contents do not support the hypothesis.
2. Since the authors did not provide any evidence to validate gut microbiota as a "key regulating factor" in malignant transformation and metastasis, the title of the article is still ambiguous.
3. No evidence (in vitro, in vivo, or clinically) has been presented linking the gut microbiome to malignant transformation and metastasis.
4. In order to prove the hypothesis, crosstalk between microbiota and epigenetic modifications was not fully demonstrated using in vitro and in vivo studies. Using tables, elaborate on the literature.
5. In figure 3. Cross-talk between microbiota and epigenetic modifications associated with gut homeostasis or disease. There is no explanation of this content in the text in relation to cancer or tumorigenic effects
6. Authors can provide detailed illustrations to substantiate the gut microbiome and their immunomodulatory and anticancer mechanisms
7. The language is very difficult to read/understand. Authors must check carefully for grammatical, stylistic, and typographical errors.
8. There were many research and review studies available related to this manuscript in Pub Med, thus there is no innovation. The manuscript was poorly written, and the title and hypothesis were controversial. The listed references are quite similar to the topic, which is in PubMed: Front Immunol. 2021 Mar 26;12:612826. doi: 10.3389/fimmu.2021.612826; Int J Mol Sci. 2019 Oct 24;20(21):5295. doi: 10.3390/ijms20215295; CA Cancer J Clin. 2017 Jul 8;67(4):326-344. doi: 10.3322/caac.21398; Sci Transl Med. 2015 Jan 21;7(271):271ps1. doi: 10.1126/scitranslmed.3010473; Eur J Clin Microbiol Infect Dis. 2017; J Cancer Res Clin Oncol. 2019 Jan;145(1):49-63. doi: 10.1007/s00432-018-2816-0. May;36(5):757-769. doi: 10.1007/s10096-016-2881-8. Front Immunol. 2020 Nov 30;11:615056. doi: 10.3389/fimmu.2020.615056; Clin Transl Oncol. 2022 Nov 14. doi: 10.1007/s12094-022-02995-5. .
Author Response
We can see that the referee is not satisfied with our changes and overall revision. However, we did our best to improve our paper by implementing all the recommendations.
- The paper has been poorly designed, and information was not gathered and organized in a meaningful manner; therefore, the present contents do not support the hypothesis.
We improve the structure of the paper and to follow the logical flow to provide information in concise and comprehensive manner.
2. Since the authors did not provide any evidence to validate gut microbiota as a "key regulating factor" in malignant transformation and metastasis, the title of the article is still ambiguous.
We disagree with this statement.
3. No evidence (in vitro, in vivo, or clinically) has been presented linking the gut microbiome to malignant transformation and metastasis.
We disagree with this statement.
4. In order to prove the hypothesis, crosstalk between microbiota and epigenetic modifications was not fully demonstrated using in vitro and in vivo studies. Using tables, elaborate on the literature.
Our paper is 10,000 words now. We cannot cover all these studies. This will lead the paper out of scope and focus.
5. In figure 3. Cross-talk between microbiota and epigenetic modifications associated with gut homeostasis or disease. There is no explanation of this content in the text in relation to cancer or tumorigenic effects
We decided to present the information in such a matter.
6. Authors can provide detailed illustrations to substantiate the gut microbiome and their immunomodulatory and anticancer mechanisms
Our focus was different.
7. The language is very difficult to read/understand. Authors must check carefully for grammatical, stylistic, and typographical errors.
We check this and other reviewers do not express this opinion.
8. There were many research and review studies available related to this manuscript in Pub Med, thus there is no innovation. The manuscript was poorly written, and the title and hypothesis were controversial. The listed references are quite similar to the topic, which is in PubMed: Front Immunol. 2021 Mar 26;12:612826. doi: 10.3389/fimmu.2021.612826; Int J Mol Sci. 2019 Oct 24;20(21):5295. doi: 10.3390/ijms20215295; CA Cancer J Clin. 2017 Jul 8;67(4):326-344. doi: 10.3322/caac.21398; Sci Transl Med. 2015 Jan 21;7(271):271ps1. doi: 10.1126/scitranslmed.3010473; Eur J Clin Microbiol Infect Dis. 2017; J Cancer Res Clin Oncol. 2019 Jan;145(1):49-63. doi: 10.1007/s00432-018-2816-0. May;36(5):757-769. doi: 10.1007/s10096-016-2881-8. Front Immunol. 2020 Nov 30;11:615056. doi: 10.3389/fimmu.2020.615056; Clin Transl Oncol. 2022 Nov 14. doi: 10.1007/s12094-022-02995-5.
We cite some of these papers. However, although they are in the same field, their focus, goals and topics.
Round 3
Reviewer 1 Report (Previous Reviewer 2)
The authors have not addressed all of my comments.
This manuscript is a resubmission of an earlier submission. The following is a list of the peer review reports and author responses from that submission.
Round 1
Reviewer 1 Report
There are a lot of issues in this manuscript to be resolved.
1.A scheme showing the mechanisms how gut microbiota alters cancer progression should be presented.
2.Introduction: how gut microbiota regulates carcinogenesis should be more detailed in the viewpoints of anti-tumor immunity, malignant transformation, spreading or metastasis of cancers.
3.L46 cancerogenesis (used by the authors ) may be changed to carcinogenesis, which is a more common term.
4.L65-67: The authors mean that Th17 pathway favors CRC development or the opposite? This should be detailed.
5.Line 198-200: what is the main species or genera producing propionate or butyrate? That should be shown.
6.L205: Butyrate rather suppresses IL-18 production by suppressing caspase 1 or NLRP3 inflammasome. Please explain the discrepancy from the authors' theory insisting anti-cancer activities of butyrate.
7.Finally what kind of manipulation of gut microbiota is useful for the inhibition of tumor progression? That should be described.
Reviewer 2 Report
The article by Boris Krastev. The study, "Gut microbiome as a key factor in oncogenesis," describes the crosstalk between microbiota and the host that may lead to significant exogenous factors for cancer. However, there are some concerns that should be addressed by the authors.
1. The paper has not been well designed. There is a need to provide tables and illustrations to substantiate the gut microbiome as a key factor in oncogenesis.
2. Significant grammar and typographical errors were found throughout the manuscript. Authors from non-English speaking countries should ensure to have their articles corrected by a native English speaker for grammatical, stylistic, and typographical errors.
3. The authors should provide their own justification and relevance of the study. The authors should explain the novelty of the study. There were over 100 papers available in PubMed related to the topic. A relevant article in the field, such as CA Cancer J Clin. 2017 Jul 8;67(4):326-344. doi: 10.3322/caac.21398; Sci Transl Med. 2015 Jan 21;7(271):271ps1. doi: 10.1126/scitranslmed.3010473; Eur J Clin Microbiol Infect Dis. 2017; J Cancer Res Clin Oncol. 2019 Jan;145(1):49-63. doi: 10.1007/s00432-018-2816-0. May;36(5):757-769. doi: 10.1007/s10096-016-2881-8. Front Immunol. 2020 Nov 30;11:615056. doi: 10.3389/fimmu.2020.615056.
4. There was no difference between the abstract and conclusion in the manuscript
Reviewer 3 Report
Authors in article entitled "Gut microbiome as a key factor in oncogenesis" tried to evaluate correlations between gut microbiota and oncogenesis. The scientific topic is interesting and worth being investigated. Unfortunatelly, presented review only perfunctorily assesses this important issue.
The title of article is misleading, since authors did not provide sufficient evidences to justify gut microbiota as a "key factor" in oncogenesis with presented references.
Additonally, authors focused mainly on gut microbiota in way that its actions affected certain immunological mechanisms that were established to take part in oncogenesis. Direct effects of gut microbiota are lacking and there are more speculations in this area.
Moreover, when it comes to gut microbiota-derived metabolites authors focused only of SCFA and bile acids. This review should have included actions of indoles, TMA, TMAO, hydrogen sulfide etc. to better justify importance of gut microbiota and its metabolites in tumor development and prognosis in oncological patients to broaden this topic by adding more clinical approach.
Since scientific recognition of IJMS increases significantly each year leading to current IF of 6,208 regrettably I cannot provide positive review of presented manuscript.